# Ten-Minute Physical Activity Breaks Improve Attention and Executive Functions in Healthcare Workers

**DOI:** 10.3390/jfmk9020102

**Published:** 2024-06-12

**Authors:** Francesco Fischetti, Ilaria Pepe, Gianpiero Greco, Maurizio Ranieri, Luca Poli, Stefania Cataldi, Luigi Vimercati

**Affiliations:** 1Department of Translational Biomedicine and Neuroscience (DiBraiN), University of Study of Bari, 70124 Bari, Italy; francesco.fischetti@uniba.it (F.F.); ilaria.pepe@uniba.it (I.P.); maurizio.ranieri@uniba.it (M.R.); luca.poli@uniba.it (L.P.); 2Section of Occupational Medicine, Interdisciplinary Department of Medicine, University of Study of Bari, 70124 Bari, Italy; luigi.vimercati@uniba.it

**Keywords:** outdoor physical activity, exergames, attentional/executive levels, cognitive functions, healthcare employees

## Abstract

Occupational health is a major problem in modern work environments. Physical activity breaks (PABs), short exercise periods delivered during working hours, incorporating exergames or outdoor activities, have emerged as a novel approach that could be used to improve work efficiency and workplace wellbeing. Therefore, this study aimed to investigate the impact of PABs on attention levels and executive functions in healthcare workers. A total of 27 healthcare workers (M = 14, W = 13; 49.55 ± 12.46 years), after 4 h of work, randomly performed one of three 10 min conditions weekly in a counterbalanced order: No Physical Activity Break (NPAB); Outdoor Physical Activity Break (OPAB); Physical Activity Break with Exergame (PABEx). After the conditions, executive functions and selective attention were assessed by the Stroop Color and Word Test (SCWT), and the Trail Making A,B test (TMT A,B), respectively. Significant differences between OPAB and NPAB as well as between PABEx and NPAB in the TMT-A test χ2(2) = 44.66 (*p* < 0.001) and TMT-B test χ2(2) = 48.67 (*p* < 0.001) were found, respectively. TMT-A and SCWT interference/time scores of the PABEx and OPAB conditions were significantly lower than those of NPAB (*p* < 0.001). In the SCWT interference/error score, no significant difference was found between the PABEx and NPAB (*p* > 0.05), but the score was statistically lower in the OPAB condition than PABEx (*p* = 0.001) and PABEx condition compared to OPAB for TMT-A (*p* = 0.001). Findings showed that the OPAB and PABEx conditions are effective in improving selective attention and executive functions in healthcare workers. Employers can foster a healthier and more productive workforce by promoting a culture of movement and prioritizing employee health, which in turn can enhance patient care outcomes.

## 1. Introduction

Sedentary lifestyles and physical inactivity pose significant public health concerns due to their association with heightened risks of cardiovascular disease, metabolic disorders, premature mortality, and mental disorders, as well as being negatively related to both mental health and cognition [1,2,3,4,5,6,7,8].

Work occupies a significant portion of individuals’ lives, particularly healthcare professionals, with the prevalence of physical inactivity ranging from 34.8% to 87.8% based on workplace settings [9,10]. At the same time, during their work shifts, some healthcare workers (e.g., nurses) face physical requirements that are not proportional to their physical fitness, thus experiencing musculoskeletal issues (e.g., back pain) [11].

Given the significant quantity of time spent at work, particularly among healthcare professionals, it is crucial to remember that individuals can attain high levels of physical activity while also engaging in prolonged sedentary behaviors. Sedentary behaviors refer to any waking activity with a low energy expenditure (≤1.5 metabolic equivalents) when sitting, reclining, or lying down. Sedentary behavior over an extended period is linked to an increased risk of chronic illnesses and a decline in mental health [12]. Most of the research in this field has been on common mood disorders, like depression [3,5,13,14]; other facets of mental health, such cognitive function, have not been studied. The activities of the human mind and the mental processes involved in processing information from the environment, reasoning, thinking, solving problems, and making decisions are collectively referred to as cognition [15,16].

Healthcare workers’ physical inactivity is a concern due to adverse working conditions, work-related stress, and excessive sitting, posing health and safety risks as well as promoting burnout, fatigue [17,18,19], and cognitive impairment [7,20,21].

Furthermore, physical activity benefits extend to brain health and cognitive functions [22,23,24], enhancing cognitive strategies for effective task performance [25].

Given the physical and cognitive demands of occupational tasks, workplace exercise emerges as a promising intervention to enhance physical, physiological, and cognitive capacities [26,27]. However, while the effects of working weekends, holidays, and end-of-day breaks have been extensively studied, the impact of active day breaks has received less attention [28,29]. The concept of an “Active Break” (AB) has been introduced, showing potential benefits in reducing musculoskeletal pain and improving work quality, efficiency, and productivity [30]. Studies on ABs indicate their usefulness in reducing fatigue as well as enhancing attention and executive functions [31].

Executive functions, crucial for daily tasks, include inhibitory control, working memory, cognitive flexibility, planning, and reasoning [32]. They are essential for success in various domains, including the workplace [33]. Deterioration of cognitive function, incapacity to solve problems, and low productivity at work are linked to low central vascular pressure [34]. Long periods of sitting are associated with vascular dysfunction, although the efficacy of active breaks in improving regional vascular outcomes remains inconclusive [35]. Research suggests that ABs lasting 5–10 min are more effective than longer durations [36,37,38].

A complex theoretical framework including insights from the fields of neuroscientific, developmental, and embodied cognition supports the effectiveness of physical activity breaks (PABs) [39,40,41,42]. Exercise-induced neurogenesis, angiogenesis, improved brain metabolism, and higher catecholamine neurotransmission are some of the physiological processes that have been proposed to explain these beneficial benefits [43,44,45]. Moreover, comprehending the advantages of PABs requires a knowledge of the notion of cognitive engagement (CE), which denotes the mental effort necessary to do complex tasks [46,47,48]. According to the “cognitive stimulation hypothesis”, engaging in physically demanding cognitive tasks activates brain regions linked to higher-order cognitive functions, which enhances executive functions [39,49,50]. Under the framework of embodied cognition, which highlights the influence of the body on the development of the mind, the cognitive advantages of PABs can also be recognized [51,52]. This viewpoint argues that physical activities that benefit the body and mind can improve cognitive performance [53,54,55]. For example, movement- and coordination-intensive exercises, including some types of exercise, can activate brain areas related to cognitive processing and motor control [56,57,58].

Additionally, exposure to natural environments and physical activity in such settings positively impacts wellbeing and cognitive functions [59]. Outdoor physical activity involves engaging in physical exercises in natural environments, such as parks, trails, or open spaces. It combines the benefits of physical exercise with the therapeutic effects of exposure to nature, including improved mood, reduced stress, and enhanced cognitive functions [60]. Activities like walking in nature can enhance mental health and cognitive performance by reducing stress and improving mood and attention [61]. Virtual reality-enhanced aerobic exercises show promise in enhancing cognitive function and confidence [62]. ABs using exergames are feasible and improve cognitive performance [63]. Active video games, or exergames, are interactive entertainment formats that combine mentally and physically taxing tasks while encouraging physical exercise through player participation [64]. The introduction of ABs in the workplace could increase health awareness, boost physical activity, and promote workers’ health and wellbeing [38].

However, the influence and types of workplace exercise on cognitive performance among healthcare workers remain underexplored [65]. Therefore, this study aimed to investigate the impact of PAB (physical activity break) interventions on the attention and cognitive functions of healthcare workers. It was hypothesized that short periods of outdoor exercise or exergames during work hours would enhance workers’ selective attention and executive functions.

## 2. Materials and Methods

### 2.1. Study Design and Setting

A randomized controlled crossover study (within-subjects repeated-measures design) was conducted between January and February 2024 to investigate the effectiveness of PABs on levels of attention and executive functions. The intervention lasted three weeks, and at the end of each protocol session, attention and executive functions were evaluated in the same setting. Cognitive assessments were performed by a research psychologist that was blinded to treatment allocations. After 4 h of work, the participants were assigned to three conditions in a random and counterbalanced order weekly: (1) No Physical Activity Break (NPAB), which was the control condition; (2) Outdoor Physical Activity Break (OPAB), which involved outdoor walking exercises; and (3) Physical Activity Break with Exergame (PABEx), which included virtual walking exercises as a break activity.

To minimize potential circadian effects, the three conditions and all assessments were performed at the same time of day, between 9:00 and 12:00, with a one-week washout period between trials. This time interval was chosen to reduce the effect of external temperatures on the OUT session. Subjects were also asked to refrain from engaging in strenuous physical activity or consuming caffeine for the 24 h preceding each condition and during data collection.

### 2.2. Participants

This study’s target sample consisted of healthcare workers from the University Hospital of Bari (Italy), as part of the University of Bari’s Horizon Seeds S70 project with the research topic “Wellbeing of healthcare workers and physical exercise”.

To establish the sample size needed for this study, an a priori power analysis [66] with an assumed type I error of 0.05 and a type II error rate of 0.20 (80% statistical power) was calculated. This revealed that 15 participants in total would be sufficient to observe medium effect sizes “within-subjects”. Twenty-seven healthcare workers (M = 14, W = 13; mean age: 49.55 ± 12.46 years) provided written consent and were informed about this study’s purpose and experimental procedures. Subsequently, each participant’s suitability was assessed using predefined criteria. The study participants were recruited based on the following criteria: (1) men and women aged 18 to 70 years old; (2) contract or full-time workers during the study period; and (3) ability to stand and exercise. The exclusion criteria included (1) symptoms or signs of musculoskeletal disorders or other serious injuries to the lower extremities, (2) the presence of acute or chronic diseases, and (3) failure to comply with the study protocol. The procedures followed were in accordance with the ethical standards of the Helsinki Declaration and approved by the Ethics Committee of Bari University (protocol code 0030611|28 March 2023).

### 2.3. Physical Activity Break Interventions

Participants were asked to refrain from engaging in any strenuous activity for at least 30 min before the experiment. Following 4 h of work, each participant performed the three weekly assigned conditions for ten minutes in a random and counterbalanced order: (1) No Physical Activity Break (NPAB); (2) Outdoor Physical Activity Break (OPAB); (3) Physical Activity Break with Exergame (PABEx). The 10 min passive workplace break functioned as the No-Physical Activity Break (NPAB). The task involved stopping work while keeping the workstation in place.

Each Outdoor Physical Active Break (OPAB) session lasted 10 min, divided into 3 phases: Warm-up: aimed to increase heart rate, improve muscular blood flow, and prepare major joints for the subsequent work phase (2 min duration). Main activity (Outdoor Walk): consisted of a low-/moderate-intensity walk in the outdoor work environment (6 min duration, predetermined speed of 4.5 km/h) [67,68]. Cooldown: consisted of relaxation exercises and static stretching (2 min duration).

Each PAB session with Exergame (PABEx) lasted 10 min, divided into 3 phases: Warm-up: aimed to increase heart rate, improve muscular blood flow, and prepare major joints for the subsequent work phase (2 min duration). Main activity (Exergame): through the immersive virtual reality of this system (Homing^®^, Tecnobody, Bergamo, Italy), it was possible to replicate movements in virtual environments and provide visual and auditory feedback necessary to correct performance. This 6 min phase involved using one of the interactive activities available that simulated a low-/moderate-intensity walk at a predetermined speed of 4.5 km/h [67,68]. Cooldown: consisted of relaxation exercises and static stretching (2 min duration).

### 2.4. Cognitive Outcomes

#### 2.4.1. Selective Attention Assessment

The Trail Making Test (TMT), a neuropsychological assessment instrument, was administered to the participants. It measures participants’ divided attention/attentional shifting and visual processing speed [69]. The TMT is composed of two parts. Part A instructs participants to connect 25 scattered dots containing numbers 1 through 25 in a sequential order. The time required to complete Part A (in seconds) is an indicator of visual processing speed. Part B of the test requires participants to connect numbered and lettered circles in an alternating sequence of numbers and letters, progressing alphabetically and numerically. The time required to complete Part B (in seconds) is a measure of divided attention. In both parts of the test, participants aim to complete the sequence as quickly as possible, and their score reflects the time it takes to complete this task (in seconds). The TMT B-A score (in seconds) is regarded as a refined measure of divided attention, unaffected by visual processing speed. This derived score is frequently used to identify the executive efficiency associated with prefrontal activation [70]. Strong test–retest reliability is shown by baseline scores and the derived TMT B-A score (Part A, r = 0.80; Part B, r = 0.81; Part B-A, r = 0.70). Age and educational level adjustments are made to the raw scores from Part A (TMT-A score), Part B (TMT-B score), and TMT B-A. Deficit is indicated by scores of 127 s or more for Part A, 294 s for Part B, and 163 s for B-A.

#### 2.4.2. Executive Functions Assessment

The neuropsychological tool used to evaluate executive functions related to the efficiency of the frontal lobe, cognitive flexibility, and sensitivity to interference was the Stroop Color and Word Test (SCWT). There are three trials in the test. In the first trial, the participant must read three lists of color names—green, red, and blue—as quickly as possible and in a random order. The second trial involves the subject reading three lists of colored dots (green, red, and blue) as quickly as possible in a random order. The third trial involves having the subject read three lists of words written in a different color (for example, “RED” written in blue ink) in any combination and in any order. In this incongruent condition, the examinee must name the color of the ink rather than read the word. The final trial assesses sensitivity to interference (that is, the ability to inhibit an automatic response). The test produces two results: sensitivity to interference related to the time spent completing the trials (interference/time) and sensitivity to interference related to committed errors (interference/errors). Raw scores (performances in seconds) were converted into age- and education-appropriate scores. The test has adequate psychometric properties [71]. The pathological cut-offs for interference/error and interference/time comprised scores of ≥4.25 and ≥36.92, respectively.

### 2.5. Statistical Analysis

Statistical analyses were performed with IBM SPSS Statistics for Windows, Version 26.0 (Armonk, NY, USA: IBM Corp.) [72]. Descriptive data were presented as a frequency distribution, with the mean and standard deviation for each cognitive outcome. A nonparametric Friedman test of differences among repeated measures due to non-normality was used to examine the effects of PABs on attentional/executive functions in the three conditions. The normality of the score distribution was first tested using the Shapiro–Wilk test. Pairwise comparisons between conditions were made using the Wilcoxon signed ranks test with a Bonferroni correction. The statistical significance was set a priori at *p* ≤ 0.05.

## 3. Results

The twenty-seven participants in this study were exposed to all three experimental conditions (NAPB, OPAB, and PABEx). Table 1 shows the mean scores for continuous cognitive variables as well as percentages for the same qualitative outcomes across the three conditions.

### 3.1. Trail Making Test (TMT)

Friedman’s non-parametric tests of differences between repeated measures of cognitive outcomes produced significant effects for visual processing speed (TMT-A = χ2(2) = 44.66, (*p* < 0.001), divided attention (TMTB = χ2(2) = 48.67, (*p* < 0.001), and executive functions (Stroop time/interference = χ2(2) = 40.67, *p* < 0.001) and Stroop error/interference = χ2(2) = 32.8, (*p* < 0.001). Bonferroni-corrected Wilcoxon signed rank tests showed significant differences between PABEx and NPAB conditions, as well as OPAB and NPAB (PABEx vs. NPAB: Z = 6.53, Adj. *p* < 0.001, OPAB vs. NPAB: Z = 4.49, Adj. *p* < 0.001), for TMT-A. There were no significant differences between PABEx and OPAB (Z = 2.04; Adj. *p* = 0.124). TMT-B revealed statistically significant differences in all comparisons. All statistically significant differences indicated a lower TMT-B score in the PABEx and OPAB conditions compared to NPAB (PABEx vs. NPAB: Z = 6.94, Adj. *p* < 0.001, OPAB vs. NPAB: Z = 4.08, Adj. *p* < 0.001) and in the PABEx condition compared to OPAB (Z = 2.86, Adj. *p* = 0.013) (see Table 2).

### 3.2. Stroop Color and Word Test (SCWT)

Executive functions, measured as sensitivity to time-related interference, were significantly lower in the OPAB and PABEx conditions compared to the NAPB (PABEx vs. NAPB: Z = 5.71, Adj. *p* < 0.001, OPAB vs. NAPB: Z = 5.31, Adj. *p* < 0.001) but not between the OPAB and PABEx (Z = 0.41, Adj. *p* = 1.00). However, a statistically significant difference was found for the sensitivity to interference associated with errors between the OPAB and PABEx conditions (Z = −4.08, Adj. *p* < 0.001). The SCWT interference/error score was not significantly different in the PABEx condition compared to the NAPB (Z = 1.02, Adj. *p* = 0.992) but was statistically lower in the OPAB condition compared to NPAB (Z = 5.10; Adj. *p* = 0.001) (see Table 2).

## 4. Discussion

This study aimed to assess the effectiveness of 10 min PABs interventions on improving healthcare workers’ attentional and executive functions. Its specific goal was to investigate the impact of work breaks facilitated by PABs taken outdoor or virtually on cognitive functions in comparison with a control group. Based on the hypothesis that short sessions of PABs outdoors and exergames would produce more significant improvements in cognitive performance than interruptions without physical movement, the results of OPAB and PABEx demonstrated improved performance in the Stroop and Trail Making Test A,B. This suggests that brief interruptions featuring active physical breaks, lasting only ten minutes, exert an acute positive influence on cognitive function among healthcare workers, effectively enhancing it. These findings align with prior research, indicating the acute cognitive benefits associated with traditional outdoor exercise and the integration of exergames [73,74,75].

Previous research has demonstrated transient cognitive enhancements in working memory [47], executive functions [76], attention, and concentration [77,78], alongside the well-established cognitive advantages associated with regular exercise. Some studies indicate that just 10 min of exercise can elicit immediate cognitive effects, a finding consistent with the methodology and results of our study [36,37,38,79]. Through the adoption of three 10 min active breaks daily, individuals can attain the physical benefits equivalent to 30 min of uninterrupted physical activity [80,81], thereby augmenting mood, attention span, and concentration throughout the day. However, the exercise environment may be just as important as the exercise itself, including duration and intensity [82,83].

We discovered an improvement in attentional indices in the Exergame group but not in executive functions, where the OPAB group performed better. All other cognitive indices (i.e., TMT-A, and SCWT interference/time) showed no statistically significant differences between the conditions, indicating neurophysiological equivalence. There is substantial evidence that both outdoor physical activity and exergames cause neuroplastic changes in the brain, such as increased synaptic connectivity, neurogenesis, and neurotransmitter levels [84,85]. These neuroplastic changes, regardless of the activity, can help to improve cognitive functions (such as attention and executive function). Neuroimaging studies have shown that both outdoor exercise and exergames activate similar brain regions associated with attention, such as the prefrontal cortex, anterior cingulate cortex, and parietal cortex [86,87]. Similar brain activation patterns during these activities may explain why there are no differences in attentional outcomes. Both outdoor physical activity and exergaming can raise arousal and alertness levels, which are linked to attentional processes [88,89]. Individuals’ neurophysiological responses to various types of physical activity may differ depending on their fitness level, cognitive reserve, and genetic predispositions [90]. The differences in cognitive outcomes between active breaks outside and those with games may be obscured by this individual variability. Moreover, an investigation of the long-term or chronic effects of active breaks might be required to detect meaningful differences between the two interventions as this study was predicated on the acute effects of outdoor breaks and exergames on attention [91]. These neurophysiological considerations make it clear that the lack of differences in some of the tested cognitive parameters between active outdoor breaks and exergames among carers may be due to the acute effects of physical activity on attentional processes and overlapping neural mechanisms.

TMT-B scores revealed statistically significant differences between the Outdoor Physical Activity Break (OPAB) condition and the Physical Activity Break with Exergaming (PABEx) condition, whereas exercise-related cognitive performance was better in the OPAB group. Exergaming improved participants’ divided attention. This is explained by the intrinsic features of exergaming, in which participants process multiple visual and auditory cues at the same time while moving physically [92,93]. Exergaming’s multitasking aspect makes it difficult to allocate resources effectively, thereby training participants’ divided attention capacity [94]. Furthermore, exergame modes frequently feature dynamic stimuli and visually complex environments that require quick processing, influencing visual processing [95].

According to research, exergaming increases activity in brain regions associated with attention and visual processing, such as the prefrontal cortex and retinal cortex [96]. These neurophysiological changes could explain the observed cognitive improvements associated with exergaming. The exergame protocol included cues designed to train divided attention and working memory. Video games that teach how to manipulate multiple sources of information have been shown to improve cognitive flexibility [97]. As a result, physical exercise could be performed in a cognitively stimulating environment to maximize cognitive benefits [98,99].

When comparing the outdoor and exergame conditions, a contradictory result was found for a component of executive functions. An increasing body of research [100,101] underscores the beneficial effects of natural environments on mental and cognitive health. Given this, it is reasonable to expect that the combined influence of nature and physical activity would lead to an overall enhancement of cognitive capacities, as reflected in our study’s findings concerning executive functions.

Evidence from various studies [102,103,104,105] supports the notion that engaging in exercise within natural outdoor settings confers greater benefits to the brain compared to indoor exercise. Notably, extensive outdoor exercise has been shown to augment prefrontal cortex-dependent executive functions encompassing working memory, attention, and inhibitory control. For instance, Bailey et al. [103] reported that individuals who engaged in outdoor walking within natural environments exhibited significantly improved performance compared to those who walked indoors on cognitive tasks such as the Stroop task, which assesses one’s cognitive processing speed and inhibitory control.

Recent studies highlight the positive effects of outdoor physical activity on cognitive processes, especially attentional resilience, and inhibitory control. Long-term exposure to natural settings, which are defined by green and blue spaces, has been linked to improved inhibitory control, a cognitive function that is essential for focusing and sifting through distractions. Immersion in natural environments improves impulse control and self-regulation because they are peaceful and captivating, according to a review by Hansen et al. [106]. Furthermore, research has shown that engaging in outdoor exercise enhances attentional control by reducing one’s sensitivity to distractions [107]. According to Rogerson et al. [108], practicing mindfulness in natural environments can enhance attentional resilience and decrease distractibility. This is the idea behind “green and blue mindfulness”. It was emphasized how important nature-based interventions are for improving cognitive wellbeing. Moreover, recent studies highlight the significance of sensory stimulation and a multi-faceted setting in outdoor activities [107,109]. Compared to indoor or exergaming settings, outdoor environments with varied sensory experiences better occupy attentional resources.

### Strengths and Limitations

The current study had several strengths and provided an original contribution to the knowledge in this area as it was the first to investigate the use of outdoor PABs and exergames after a 4 h work shift to improve attentional and executive functions. The study’s innovative approach to stress management stands out due to the demanding nature of healthcare worker’s roles, which can impair cognitive abilities. This study introduced new perspectives in interventions for both mental wellbeing and performance improvements by investigating the impact of PABs, including outdoor exercise and exergaming, on cognitive functions among healthcare workers.

This study used a robust methodology, with a randomized crossover design and cluster sampling. This meticulous approach ensured the reliability of the results and validity of this study, which was possible due to the rigorous inclusion criteria, within-subjects design (taking advantage of sample homogeneity), and use of standardized cognitive tests. However, this study has several limitations: (1) The sample size was small, which may limit the findings’ generalizability. (2) This study examined the acute effects of PABs on cognitive function, with interventions lasting only 10 min. The long-term impact and sustainability of these interventions were not assessed. Future research should look at the long-term benefits of taking regular PABs. (3) There were no differences in gender examined. According to research, men and women may benefit from physical activity in different ways due to differences in hormone levels, muscle mass, and metabolic rates [110,111]. For example, studies have revealed that men may have higher increases in specific cognitive processes, such as executive functioning and spatial memory, after physical activity [112,113], whereas women may benefit more in terms of mood and affect [114,115].

Based on these findings, future studies ought to adapt PAB procedures to account for any gender differences.

## 5. Conclusions

To the best of our knowledge, this is the first study to provide new information about the benefits of short breaks, defined as PABs, on cognitive function in healthcare workers.

This study aimed to assess the effectiveness of 10 min physical activity breaks (PABs) in improving attentional and executive functions among healthcare workers. The findings demonstrate that brief, structured PABs, whether outdoors or through exergaming, can have acute positive effects on cognitive performance.

Outdoor PABs and exergames both improved attention indices, but the OPAB group outperformed the other groups in executive functions. These findings indicate that the type of intervention used can influence specific aspects of cognitive function, emphasizing the importance of tailoring interventions. Promoting PABs in the workplace is critical for improving cognitive function and wellbeing in healthcare workers. Employers can help create a healthier, more productive workforce by encouraging a movement culture and prioritizing their employee’s health, ultimately improving patient care outcomes.

In terms of levels of care, our study highlights the need for multi-faceted interventions that cater to the varying demands of healthcare settings. For primary care providers, incorporating PABs can help manage the high cognitive load associated with patient interactions and administrative tasks. Secondary care professionals, who often face intensive and specialized patient care responsibilities, can benefit from PABs to mitigate the effects of prolonged sedentary behavior and mental fatigue. Tertiary care workers, operating in highly specialized and high-pressure environments, can use PABs to enhance their cognitive resilience and maintain high levels of performance. We recommend for the implementation of personalized PAB interventions to take place at various levels of care in healthcare settings. These interventions should be adapted to the specific cognitive and physical demands of each level, resulting in a healthier, more productive workforce.

## Figures and Tables

**Table 1 jfmk-09-00102-t001:** Cognitive characteristics of participants for each condition.

Cognitive Variables	Mean	Std. Deviation
**TMT—A score**		
NPAB	84.0	11.77
OPAB	56.0	10.97
PABEx	48.78	9.47
**TMT—B score**		
NPAB	178.89	20.80
OPAB	136.56	13.70
PABEx	125.67	18.63
**SCWT interference/time score**		
NPAB	32.06	7.67
OPAB	23.10	4.74
PABEx	24.88	5.07
**SCWT interference/error score**		
NPAB	2.91	0.75
OPAB	2.02	0.28
PABEx	2.63	0.68

**Table 2 jfmk-09-00102-t002:** Cognitive changes found among the three conditions.

Cognitive Variables	NPAB	OPAB	PABEx	Test Statistic	Asymptotic Sig.
TMT-A mean rank	3.00 ^a,^***^,b,^***	1.78 ^a,^***	1.22 ^b,^***	44.66	<0.001
TMT-B mean rank	3.00 ^a,^***^,b,^***	1.89 ^a,^***^,c,^*	1.11 ^b,^***^,c,^*	48.67	<0.001
SCWT Interference-Time mean rank	3.00 ^a,^***^,b,^***	1.56 ^a,^***	1.44 ^b,^***	40.67	<0.001
SCWT interference-Error mean rank	2.56 ^a,^***	1.17 ^a,^***^,c,^***	2.28 ^c,^***	32.81	<0.001

Data are reported as mean rank. Abbreviations: NPAB, No Physical Activity Break; OPAB, Outdoor Physical Activity Break; PABEx, Physical Activity Break with Exergame; TMT-A, Trail Making Test Part A; TMT-B, Trail Making Test Part B; SCWT, the Stroop Color and Word Test; Asymptotic Sig., Asymptotic Significance. ^a^ Significant difference between NPAB and OPAB; ^b^ significant difference between NPAB and PABEx; ^c^ significant difference between OPAB and PABEx. * *p* < 0.05; *** *p* < 0.001.

## Data Availability

The data presented in this study are available upon request from the corresponding author. The data are not publicly available due to privacy restrictions.

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
