# Peer review of "Ten-Minute Physical Activity Breaks Improve Attention and Executive Functions in Healthcare Workers"

_jfmk, 2024, doi:10.3390/jfmk9020102_

Round 1
Reviewer 1 Report
Comments and Suggestions for Authors
The research presented is original and provides useful results for the knowledge of health as a social problem.
I thank the authors for the valuable exercise they carried out.
The appreciations are attached in the PDF document, so it is respectfully suggested to review each of the comments, which have the purpose of improving the manuscript generated.
A final note: the references should be reviewed in depth in order to adjust them to the journal's standards.

Reviewer 2 Report
Comments and Suggestions for Authors
Thank you very much for submitting the manuscript, which addresses an important topic in today's society. I have a few comments that may be worth considering.
#Lines 38/39: Perhaps the psychological effects could also be directly presented here, as the contribution also focuses more on this aspect.
#Lines 41-43: I find the emphasis on inactivity her very one-sided. There a many health professions (e.g., nursing) that involve a high amoung on physical activity and little inactivity. However, this activity is often detimental to health (e.g., lifting heavy weights and the resulting back problems), which is why it is important to include physical activities that conteract these negative effects. Perhaps this could be addressed.
#Lines 49-53: There is a lot of focus on the physical effects hiere, which are not the main objective of this work. Perhaps this paragraph could be shortened and more emphasis placed on the psychological effects (overall in the instruction).
#Entire introduction: The introduction lacks the theoretical background. It would be beneficial to introduce and explain a theoretical model that can elucidate the effects of active breaks on cognitive functions.
#Line 95: Were the variables also collected at the beginning of the studie (pre-test)? Could information on this be added?
#Line 128: The number 4 is sometimes written out (four) and sometimes as a digit (4) This also applies to other numbers. Please make this consistent.
#Lines 128-131: This sentence has already been written once before. Perhaps the repitition can be avoided.
#Line 138: Was the speed predetermined or chosen by each participant themselves? For untrained individuals, 4.5km/h may no longer be considered low/moderate intensit. If the speed was predetermined and not subjectively chose, it might be worth explaining why 4.5km/h was selected.
#Line 188: Please cite SPSS (https://www.ibm.com/support/pages/how-cite-ibm-spss-statistics-or-earlier-versions-spss)."
#Line 256: Is there an extra space here?
